# Inspection Application in an Industrial Environment with Collaborative Robots

**Paulo Magalhaes** [1] **and Nuno Ferreira** [2,3,4,*]

1   Europneumaq, Rua Senhora da Mestra, nº 35, 4410-511 Serzedo, Portugal; p.magalhaes@europneumaq.pt
2   Engineering Institute of Coimbra (ISEC), Polytechnic of Coimbra (IPC), Rua Pedro Nunes—Quinta da Nora, 3030-199 Coimbra, Portugal
3   GECAD—Knowledge Research Group on Intelligent Engineering and Computing for Advanced Innovation and Development, Engineering Institute of Porto (ISEP), Polytechnic Institute of Porto (IPP), 4200-465 Porto, Portugal
4   INESC TEC—Institute for Systems and Computer Engineering, Technology and Science, 4200-465 Porto, Portugal
*   Correspondence: nunomig@isec.pt

**Abstract:** In this study, we analyze the potential of collaborative robotics in automated quality inspections in the industry. The development of a solution integrating an industrial vision system allowed evaluating the performance of collaborative robots in a real case. The use of these tools allows reducing quality defects as well as costs in the manufacturing process. In this system, image processing methods use resources based on depth and surface measurements with high precision. The system fully processes a panel, observing the state of the surface to detect any potential defect in the panels produced to increase the quality of production.

**Keywords:** inspection; industrial quality control; vision systems; collaborative robotics

## 1. Introduction

The use of inspection systems can prevent defective products and ensure quality throughout the manufacturing process. In manual assembly lines, operators carry out inspections, and this process is time-consuming and quite expensive. However, these inspections are unreliable, for example, due to lack of training, fatigue, or unclear specifications [1]. Therefore, inspection methods for vision systems are used in industry and various domains, such as the food industry and safety applications [2].

Vision systems are recommended for critical inspections if the correct image acquisition system can be implemented [3]. A current problem with most vision systems that limit their use in manual assembly lines is fixed cameras, making the system inflexible in its way of working. A fixed camera system has problems reaching inaccessible areas of the product, such as the underside, inside, and around the product to be inspected [4]. In the process of manual inspection, the mounted cameras can also interfere with the operator.

The possibility of using collaborative robots introduces flexibility into the production process, automatic quality inspections via vision, with the integration of the quality control system to the robot's control system significantly improves the performance of this type of system. Collaborative robots are becoming increasingly popular in various manufacturing processes, especially assembly, both for their flexibility and efficiency [5]. The goal of a collaborative robot is to combine repetitive performance and strength with the abilities of a human being. Several tasks can be performed by collaborative robots, their use in assembly lines, delivery of parts to be assembled, lifting heavy objects, among many other tasks. Furthermore, mounting an inspection camera on the collaborative robot and integrating the quality control software with the robot control software allows inflexibility of the vision system to be reduced. In addition, the fast-advancing continuous AI revolution has

introduced several image processing algorithms that are unquestionably very promising for industrial quality control. Therefore, the aim is to integrate the latest generation of collaborative robots to obtain a flexible and high-performance solution for automatic quality control.

Human–robot collaboration is a new trend in industrial robotics and in-service robotics as part of the industry 4.0 strategy. Robots and strategy are known by the acronym HRC (human–robot collaboration). The main objective of this innovative strategy is to build an environment for secure collaboration between humans and robots. There is an area between manual production and fully automated production where a human worker comes into contact with a machine [6]. This area has many limitations due to security restrictions. The machine can be in automatic work mode only if operating personnel are outside its working area. Collaborative robotics allows us to establish new opportunities in the cooperation between humans and machines. People can share the workspace with the robot to help with non-ergonomic, repetitive, uncomfortable, or even dangerous operations.

Through its advanced sensors, the robot can monitor its movements and limit its speed at any time or even stop performing a specific task, but significantly not to endanger its human counterpart. The possibility of using the robotic system's working space without security fences can enhance fully automated production by optimizing installation space and the costs of security barriers. From the basic principle of cobots, it is possible to increase their payload, not the load levels of industrial robots. This strategy aims to extend the different robotic applications, especially those that have not yet been automated. This type of robotic system is used by robotic assistants to improve the quality of work of human workers. In many production fields, it can be challenging to find a suitable application for collaborative robots, but demonstrating that there are many advantages, it is necessary to take advantage of the flexibility of cobots.

Considering the evolution of the state of the art, the paradigm for the development of a new inspection system is the possibility of remaining current for a long time; one of the ways to do that is the construction of simple and flexible prototypes, also the equipment used must be easy to install and have a quick configuration or parameterization so that it remains current and useful for what it was developed.

At the beginning of this article, an introduction to inspection applications with collaborative robots was made. Then, Section 2 shows the state of the art of collaborative robots in the industry. Then, in Section 3, we present the different technologies for computer vision systems in the industry, and in Section 4, the industrial prototype developed for quality inspection is presented, and, finally, we present our conclusions.

## 2. Materials and Methods

### 2.1. Collaborative Robots

Collaborative robots can, in most cases, function without the need for peripheral protection, characteristics that traditionally contribute to reduced flexibility. Furthermore, collaborative robots can be easily reprogrammable, even by workers without in-depth knowledge of programming languages for robotics [7]. For these reasons and considering the growing demand for customized products with shorter delivery times, collaborative robotics is experiencing significant growth [8]. Thus, it is expected that there will be a short-term expansion of the use of collaborative robotics aimed at new applications [9–17].

In 2008, Universal Robotics developed the first collaborative robot, called UR5, having leveraged industrial interest in applying cobots on the factory floor of various industries [15,18] and made companies such as KUKA or ABB bet on the development of cobots. In fact, since collaborative robotics allows the joint operation between humans and robots to carry out tasks in a cooperative environment, implying proximity between humans and machines, there is a set of demanding security measures that cobots must comply with to obtain international standards ISO 10218 1 and 10218 2, with emphasis on the following [15,17].

The operator and cobot do not work if the distance is less than the operator's safe distance, the equipment slows down to a stop (if the spacing is less than the safe distance and the system processes continuously and checks throughout the entire moment this condition). The ISO/TS 15066:2016 standard defines a set of good practices in implementing cobots in collaborative environments, without it being strictly necessary to stop or turn off the equipment whenever the system may interfere with human beings. According to Muller et al. [19], cobots and operators can operate in a coexistence perspective; that is, when the operator and the cobot work in the same environment but tend not to interact, when the operator and the cobot work in the same space and on the same product, and the synchronization between cobots and the operators occur at different times. Cooperation between cobot and the operators occurs when the operator and the cobot perform tasks simultaneously in the same workspace, despite performing different and collaborative tasks, and when the operator and cobot perform a task together.

Thus, collaborative robotics allows an increase in the quality of the products produced, the efficiency of production, and the improvement of operators' working conditions [20,21]. Thus, companies from different sectors of activity seek to integrate cobots into their facilities. In this context, SMEs deserve to be highlighted, due to the difficulty they present in automating their production process using traditional industrial robots, in terms of space availability and investment [22], as well as companies in sectors characterized by mass production, such as the automotive sector [15].

*2.2. Computer Vision Systems*

In terms of computer vision systems for object recognition, in the industry, in recent years, there has been a growing demand for optical sensors for three-dimensional (3D) images, which has driven the development of new, more sophisticated instruments. Three-dimensional image sensors generally work by projecting their active form or acquiring, or in passive terms, electromagnetic energy into an object, then recording the transmitted or reflected energy. In the specific domain of robotics, several approaches integrate this component with the acquisition of 3D information, allowing the machine to incorporate the perception of three-dimensional information from its surrounding space into the machine, allowing it to interact with the same space/scenario, including the manipulation of objects or even the communication with other machines. Following are some of the most used technologies in computer vision. Photogrammetry technology covers other practical challenges: imprecision, measurement speed, automation, process integration, cost performance, and sensory integration. Photogrammetry systems are highly accurate, and we find a specific set of applications in the automotive industry, such as measuring the deformation of bodies, controlling parts, adjusting tools and platforms, among others. [23]. Furthermore, integrated into a network, the photogrammetry systems provide information in 3D that enables the connection and control of processes. It is essential to highlight that photogrammetry systems can be coupled to industrial robots, allowing them to guide the execution of drilling and assembly tasks [23,24].

Time-of-Flight (ToF) technology works by emitting a beam of light to a surface, quantifying the time elapsed between the emission and reflection of light to determine the effective distance to the object. ToF sensors are very compact and light, allowing the acquisition of real-time 3D images with high resolution, used in moving arms of robotic systems. Structured light technology uses light patterns to acquire 3D images of a particular object or surface, is hardly affected by the variability associated with light, which facilitates the correct detection of objects. However, sensors and the design of considerable dimensions make it difficult to attach to the end of the robotic system. The laser triangulation technique explores the identification of parts and can also be used in different stages of welding. Typically, 2D cameras allow the detection and identification of objects, being, however, sensitive to light and clean conditions; that is, the presence of dust in the industrial environment [25], which can lead to the incorrect selection of components to moving or screwing components in the wrong places. Therefore, the integration of 3D vision systems in cobots represents

one of the options explored to face the limitations currently presented by 2D vision systems. Vision systems play a crucial role in many robotic applications, both in automated and fixed industrial operations or quality control and in more advanced tasks, such as object selection and assembly operations [26].

However, several other factors are observed in an industrial context, such as the existence of untextured surfaces and variations in angles and dimensions. These photometric variations include changes in brightness and contrast, or perspective deformations due to a change in the observer's position, moving objects, among many others, which potentialize technical challenges for which the resolution is not trivial [23]. In addition, 3D visions systems identify and quantify the distance operators are from the machine, redefine the cobot's trajectory as it approaches obstacles, and interpret human gesture orders [26].

The objective solution will involve developing a three-dimensional capture system or three-dimensional features of objects based on the exploration of three-dimensional image acquisition techniques. This system can allow the integration of different technologies to complement, on the one hand, the knowledge of the environment surrounding the robotic system and, on the other hand, guide the robotic system in its tasks/operations, namely in the identification of objects and functions of tightening of automotive components. In addition, the development of a fast and efficient artificial vision system capable of controlling and ensuring the rapid response time of cobots represents a significant challenge since the industrial sector is characterized by a reduced cadence time to maximize its productivity and incorporate such innovations into the production line and gain a unique competitive advantage over other competitors. For example, the study by Li et al. [23] culminated in the development of a cobot capable of unscrewing screws, and [26] developed a solution to remove screws from laptops, using artificial vision to locate the screws.

## 3. Industrial Prototype Implementation

The prototype was developed with a focus on quality inspection, and to evaluate the solution that integrates vision systems and collaborative robots for industrial quality inspection, we chose the collaborative robot from Doosan M0617, considered to be the appropriate size for the inspection task and the load required to support the vision system, the Doosan M0617 collaborative robot has a payload of 6 kg and a range of 1700 mm, as shown in Figure 1. Furthermore, a clear advantage of the collaborative robot is that it easily mounts a camera and integrates the vision system and the robot control system to locate objects in a specific working area.

For a human, it is usually easy to identify the object of interest, but it is considerably more challenging to develop an algorithm to do the same. Locating an object involves demanding math, subject to high complexity, a problem affected by pixel inputs. To put this in perspective, for example, a low-quality image of $256 \times 256$ pixels results in more than 1.5 million operations. This indicates that, even in low-quality images, object detection becomes quite complex. There are several methods to deal with this type of problem or detect some distinct objects in the image. During this real-world study, we noticed that object location is a complex problem, as image content can vary greatly depending on, for example, the product variant or the angle from which the image is taken.

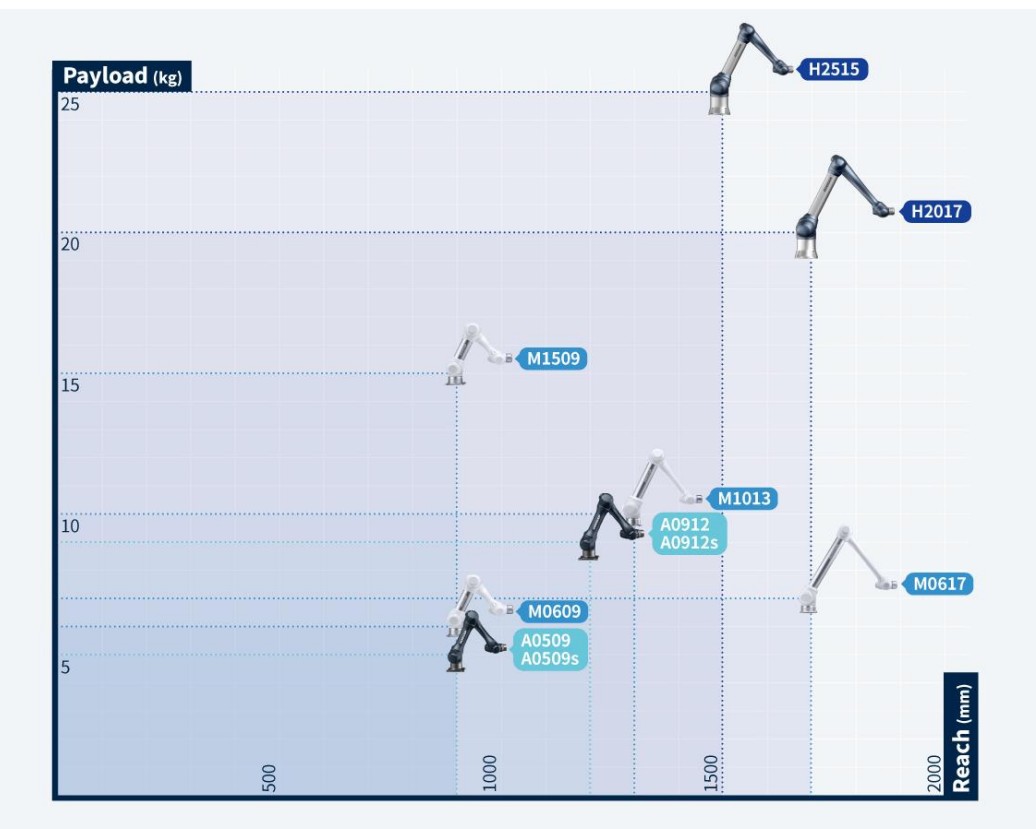

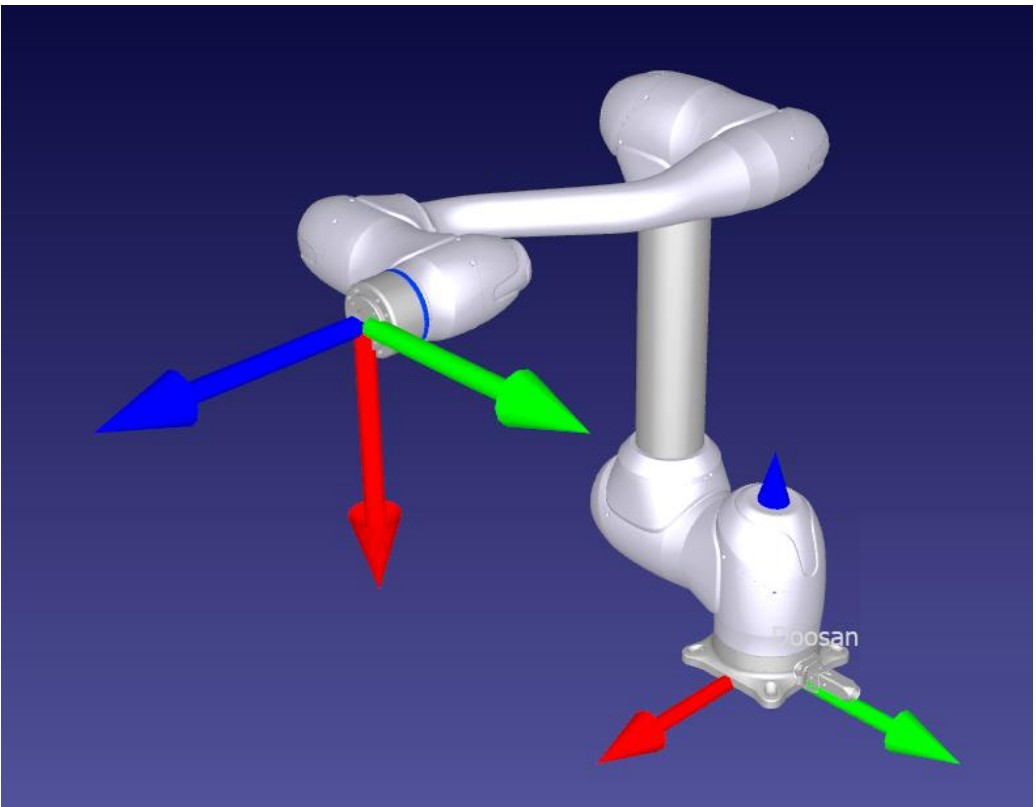

**Figure 1.** Collaborative robot from Doosan M0617.

During the implementation of the prototype, in Figure 2 and Supplementary Materials, the improvement in recognition depends on the position of the image, and, in Figure 3, for this, all the places where the images are taken were fixed for each inspected object.

Many tools allow an easy configuration of this type of application. However, if we need to send information to the robot so that it can inspect the parts, in this type of application, it is necessary to calibrate the camera so that the robot knows where the focus point is, it is necessary to calibrate the plane of the object we intend to locate. It is necessary to convert the positions of objects found in the camera coordinate system to the robot coordinate system.

Three-dimensional scanners use various techniques, such as shading, laser triangulation, structured lighting. Depending on the technique, we can obtain approximate data in seconds or extra precise data that will take longer to digitize and process. For example, in Figure 4, in our case, we chose to use the TriSpector1000 system.

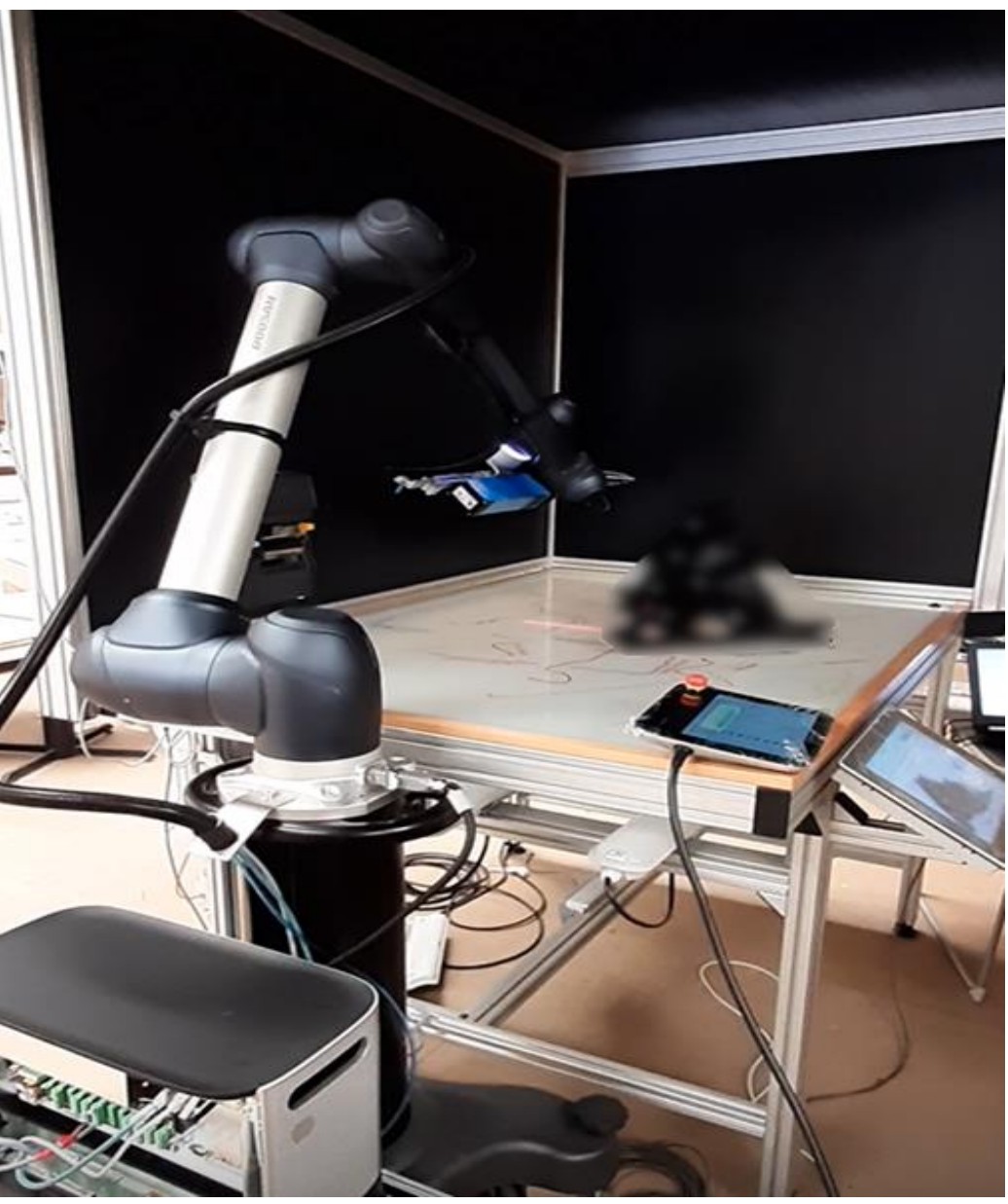

**Figure 2.** Industrial prototype for quality inspection.

TriSpector1000 builds an image by acquiring many laser line profiles of a moving object. The optimal exposure time produces a gray line in the 2D sensor image representation. The region of interest of each tool is defined and displayed as a yellow box in the image area, where objects of different shapes are located, such as edges and planes. Thus, it is possible to measure the distance or angles between objects or located features.

The tool locates a 3D shape in the image and repositions the tools according to the object's position or rotation. The Blob tool locates clusters of points within a defined height and size range, allowing the measuring of volume, area, angle, and the bounding box. The edge tool looks for edges, and the plane tool uses pixels within its region of interest to find a flat surface. The Fix Plane tool is used to define a reference plane in the field of view. The reference plane can be used as a resource for other tools. The Peak tool finds the point with the minimum or maximum height value in the region of interest. Finally, the Point tool is used to set a reference point in the field of view. This tool can be used to define point features with other tools or independently use the tool's output.

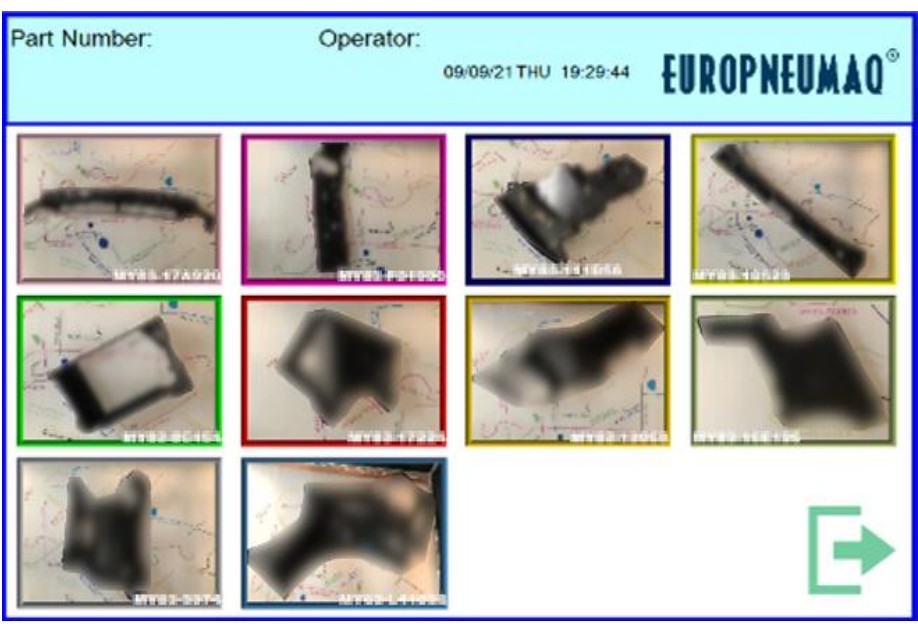

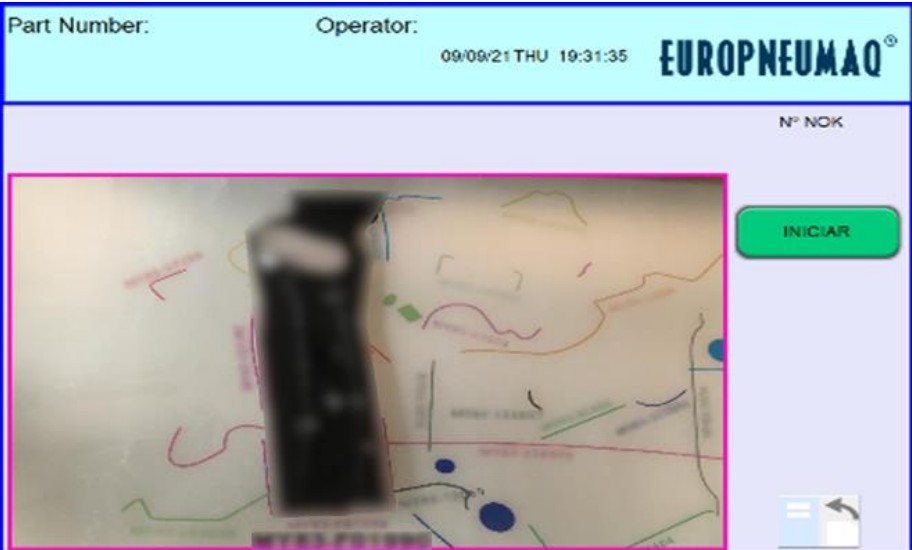

**Figure 3.** Position and orientation of the objects.

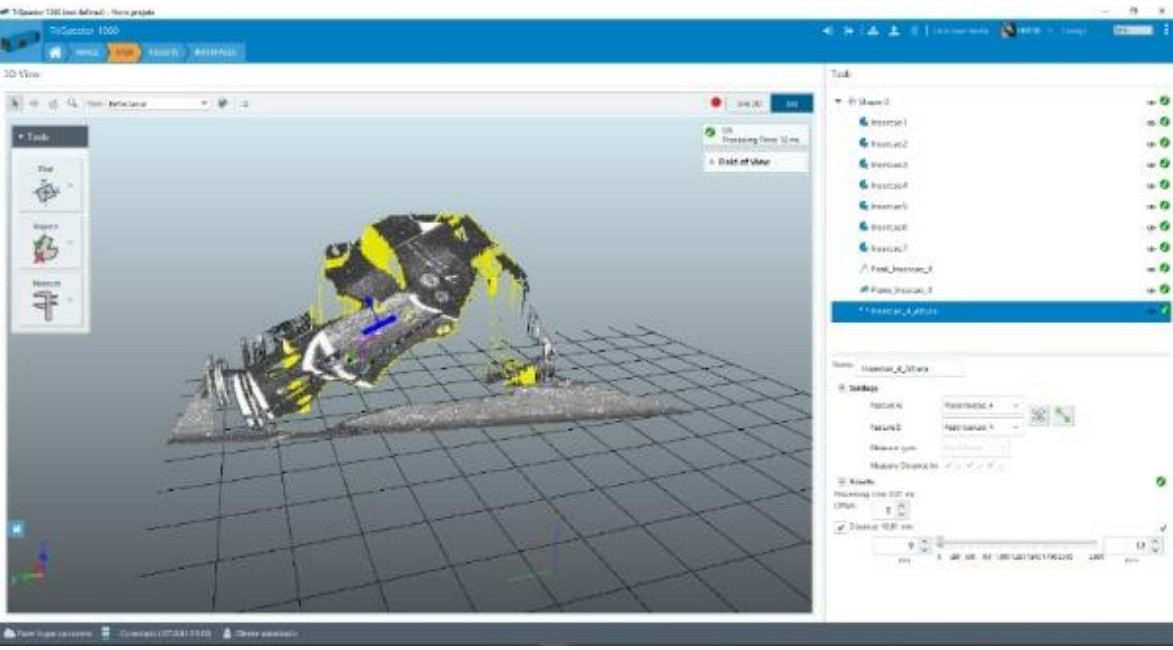

**Figure 4.** SOPAS Engineering Tool application for the TriSpector1000 system.

The Area tool calculates the surface by counting points within a defined 3D region or specified intensity range. The Distance tool measures the distance between two points on the tool. Finally, the Angle tool measures the angle between two planes. Result handling consists of methods for defining and formatting device output. The use cases range from activating a rejection mechanism to outputting results. Binary results are based on user-defined tool tolerances. Tools return results as values (e.g., area: 40 mm2, angle: 30°) or binary results (OK/Not OK).

TriSpector1000 supports a set of operators and functions for handling results. The categories of operators and functions are math, logic, and strings. In addition, TriSpector1000 has EtherNet/IP support.

As for the security system, after risk assessment, we chose to implement a security scanner that ensures that the operator is not in the space where the robot can take the camera. In this way, we avoid any shock, even if the robotic system can work at very low speeds, especially when inspecting 2 of the 16 part models to be checked. The safety laser scanner used is a Type 3, nanoScan3 (NANS3-CAAZ30AN1), range 3 m, detection angle 275°, from Sick, Figure 5.

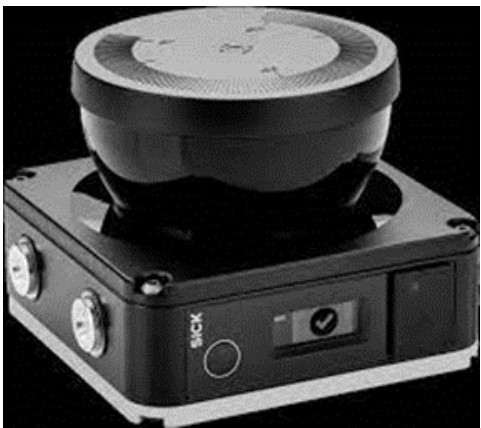

**Figure 5.** NANS3-CAAZ30AN1 safety laser scanner.

The collaborative robot was inserted into the production line, and it performed the task for which it was developed well. With efficient safety devices, the cobot increases safety in the human/robot interaction space. In addition, the software's security parameter settings are accessible by authorized and properly trained technicians and engineers, allowing for quick adaptation of the cobot in a scalable environment. Through different questions, all employees gained experience with collaborative robotics for new projects. However, the collaborative robot is still restricted to tasks with slow production cycles or low-frequency tasks.

## 4. Discussion

This article discusses the concepts of a high-performance solution for automated quality inspections in the manufacturing industry that integrates collaborative robots, leading to reduced quality defects and reduced costs in the manufacturing process. For companies, this implies that introducing human–robot solutions should ideally be preceded by an opportunity for employees to gain positive experiences with robots. As the study was in a manufacturing setting, it is not surprising that the manufacturing application area is rated slightly higher than before.

Using the software designed by Sick, Sopas, it is possible to geometrically parameterize the inspection places, the dimensions to be considered, and obtain the logical results to verify whether or not the measurements follow the pre-established values.

In the proposed application, the use of collaborative robotic systems allows an excellent interaction between the machines and the personnel who operate with them. In these inspection processes, high work rates are not required, which provides a great advantage for collaborative robots over the option of using industrial robots. In this inspection process, the operator can select different parts to be inspected, and the use of a rigorous measurement system makes it possible to perform all dimensions in detail at a high inspection speed, guaranteeing high quality and reliability of the inspected part, avoiding errors as well as preventing errors and corrections in the downstream system.

## 5. Conclusions

In the future, the development of collaborative robotics in the industry will be expressed by the close interaction of humans and robots. Under these conditions, the robot's working space intersects with a person's personal space. For an effective interaction between a human operator and a robot, it is necessary to consider production tasks. Industry 4.0 changes the structure of production processes, and humans become the center of the industrial system. Thanks to the favorable emerging conditions, collaborative robots become significantly brighter, demonstrate the benefits of reliable and secure cooperation, and increase the productivity and efficiency of task execution.

The measurements made on the parts to be inspected are validated or not, according to the rejection criteria imposed by the quality system, to guarantee the dimensions requested in the specifications for each object to be inspected. The system has been in full use in the company for more than 6 months and presents excellent results, in addition to meeting the inspection quality requirements, it is a very easy system to parameterize, which allows the inclusion of new parts for inspection by the operators specialized in this task.

**Supplementary Materials:** The following are available online at https://www.mdpi.com/article/10.3390/automation3020013/s1, Video S1: Industrial prototype for quality inspection.

**Author Contributions:** Conceptualization, P.M. and N.F.; methodology, P.M. and N.F.; software, P.M. and N.F.; validation, P.M. and N.F.; formal analysis, P.M. and N.F.; investigation, P.M. and N.F.; resources, P.M. and N.F.; data curation, P.M. and N.F.; writing—original draft preparation, P.M. and N.F.; writing—review and editing, P.M. and N.F.; visualization, P.M. and N.F.; supervision, P.M. and N.F.; project administration, P.M. and N.F. All authors have read and agreed to the published version of the manuscript.

**Funding:** This research received no external funding.

**Institutional Review Board Statement:** Not applicable.

**Informed Consent Statement:** Not applicable.

**Data Availability Statement:** Not applicable.

**Conflicts of Interest:** The authors declare no conflict of interest.

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
