# Peer review of "Inspection Application in an Industrial Environment with Collaborative Robots"

_2673-4052, doi:10.3390/automation3020013_

Round 1

Reviewer 1 Report

This paper reviews the methods of collaborative robots and computer vision systems in the industry, and the industrial prototype developed for quality inspection. However, there are some points that the authors could eventually consider. They are:

  1. In the Introduction, it is suggested to be further analyzed to highlight the finds or motivations of the manuscript.
  2. The literature review is not up-to-date.
  3. The technology and theory for computer vision systems used in this research should be explained in detail.
  4. Group related expresseswith more emphasis the difference with the proposed approach.
  5. The authors claim “in section three, we present the different...”, but the collaborative robots and computer vision systems are all in Section 2.
  6. What is the novelty of your paper?
  7. The authors should describe the implementation in more detail.
  8. The results of the implementation should be added.
  9. The authors claimed “image processing methods use resources based on depth and surface measurements with high precision. The system fully processes a panel, observing the state of the surface to detect any potential defect in the panels produced to increase the quality of production”, but there are no experimental results to support this conclusion.

Author Response

This paper reviews the methods of collaborative robots and computer vision systems in the industry, and the industrial prototype developed for quality inspection. However, there are some points that the authors could eventually consider. They are:

  1. In the Introduction, it is suggested to be further analyzed to highlight the findings or motivations of the manuscript.

Line 73 to line 77

Considering the evolution of the state of the art, the paradigm for the development of a new inspection system is the possibility of remaining current for a long time, one of the ways to do that is the construction of simple and flexible prototypes, also the equipment used must be easy to install and have a quick configuration or parameterization so that it remains current and useful for what it was developed.

  1. The literature review is not up-to-date.

We made the up-to-date 10 to 16 references.

  1. Soini, A., (2001). Machine vision technology take-up in industrial applications., IEEE.
  2. Brosnan, T. & Sun, D.-W., (2004). Improving quality inspection of food products by computer vision-a review. Dublin, Elsevier.
  3. Vithu, P. & Moses, J., (2016). Machine vision system for food grain quality evaluation: A review. Trends in food science & technology, Volume 56, pp. 13-20.
  4. Davies, R. E., (2012). Machine Vision: Theory, Algorithms, Practicalities. 4th ed. London: Elsevier.
  5. Michalos, G. et al., (2014). ROBO-PARTNER: Seamless Human-Robot Cooperation for Intelligent, Flexible and Safe Operations in the Assembly Factories of the Future. Procedia CIRP.
  6. Thiemermann, S, (2005). Direkte Mensch-Roboter-Kooperation in der Kleinteilemontage mit einem SCARA-Roboter. Dissertation, University of Stuttgart.
  7. Pentikainen, T et al, (2016). How to make collaborative robot programming easier. Teknisk- naturvetenskaplig fakultet UTH-enheten Platbrood, F e Görnemann,
  8. Platbrood, F e Görnemann, O, (2018). Safe robotics – A segurança em sistemas robóticos colaborativos. Sick – Sensor Intelligence.
  9. Vagas, M et al, (2015). The concept of human-robot cooperation. Transfer inovácií 32/2015.
  10. José Saenz & Norbert Elkmann & Olivier Gibaru & Pedro Neto, (2018). Survey of methods for design of collaborative robotics applications-Why safety is a barrier to more widespread robotics uptake, ICMRE 2018: 4th International Conference on Mechatronics and Robotics EngineeringFebruary, Pages 95–101, https://doi.org/10.1145/3191477.3191507
  11. V. Villani, (2018). Survey on human–robot collaboration in industrial settings: safety, intuitive interfaces, and applications, Mechatronics
  12. Federico Vicentini, (2020). Terminology in safety of collaborative robotics, Robotics and Computer-Integrated Manufacturing, Volume 63, https://doi.org/10.1016/j.rcim.2019.101921
  13. Luca Gualtieri & Erwin Rauch & Renato Vidoni, (2021). Emerging research fields in safety and ergonomics in industrial collaborative robotics: A systematic literature review” Robotics and Computer-Integrated Manufacturing, https://doi.org/10.1016/j.rcim.2020.101998
  14. Federico Vicentini & Mehrnoosh & Askarpour & Matteo G. Rossi & Dino Mandrioli, (2020). Safety Assessment of Collaborative Robotics Through Automated Formal Verification, IEEE Transactions on Robotics, Volume: 36, Issue: 1, DOI: 10.1109/TRO.2019.2937471
  15. S. El Zaatari, (2019). Cobot programming for collaborative industrial tasks: an overview Rob. Auton. Syst. (2019), https://doi.org/10.1016/j.robot.2019.03.003
  16. F. Sherwani & Muhammad Mujtaba Asad & B.S.K.K. Ibrahim, (2020). Collaborative Robots and Industrial Revolution 4.0 (IR 4.0), International Conference on Emerging Trends in Smart Technologies (ICETST), DOI: 10.1109/ICETST49965.2020.9080724
  17. Eloise Matheson, Riccardo Minto, Emanuele G. G. Zampieri, Maurizio Faccio and Giulio Rosati, (2019), Human–Robotic Collaboration in Manufacturing Applications: A Review, Robotics 2019, 8, 100.
  18. Universal Robotics, Our history, (2018), https://www.universal-robotics.com/about-universal-robotics/our-history.
  19. Shirine El Zaatari, Mohamed Marei, Weidong Li, Zahid Usman, (2019), Cobot programming for collaborative industrial tasks: An overview, Robotics, and Autonomous Systems, Volume 116, Pages 162-180.
  20. Müller, R., Vette, M., Geenen, A., (2017) Skill-based dynamic task allocation in Human-Robotic-Cooperation with the example of welding application. Procedia Manuf., 11, 13–21.
  21. R. Muller, M. Vette, O. Mailahn, (2016), Process-oriented task assignment for assembly processes with human–robotics interaction, Proc. CIRP 44, 210–215.
  22. L. Peternel, W. Kim, J. Babic, A. Ajoudani, (2017), Towards ergonomic control of human–robotics co-manipulation and handover, in 2017 IEEE-RAS 17th International Conference on Humanoid Robotics (Humanoids), IEEE.
  23. European Commission, (2018), Periodic Reporting for period 1 - colrobotics (collaborative robotics for assembly and kitting in smart manufacturing), Tech. Rep.
  24. Pérez, L.; Rodríguez, Í.; Rodríguez, N.; Usamentiaga, R.; García, D.F. Robotics Guidance Using Machine Vision Techniques in Industrial Environments: A Comparative Review. Sensors (2016), 16, 335.
  25. Clarke, T.; Wang, X. The control of a robotics end-effector using photogrammetry. Int. Arch. Photogramm. Remote Sem. (2000), 33, 137–142.
  26. Roni-Jussi Halme, Minna Lanz, Joni Kämäräinen, Roel Pieters, Jyrki Latokartano, Antti Hietanen, (2018), Review of vision-based safety systems for human-robot collaboration, Procedia CIRP, Volume 72, Pages 111-116.
  27. A. Tellaeche, I. Maurtua, and A. Ibarguren, "Use of machine vision in collaborative robotics: An industrial case.," 2016 IEEE 21st International Conference on Emerging Technologies and Factory Automation (ETFA), Berlin, 2016, pp. 1-6.
  28. The technology and theory for computer vision systems used in this research should be explained in detail.

Line 286-288

Using the Sopas software designed by Sick, it is possible to parameterize the inspection place geometrically, the dimensions to be taken into account, and obtain the logical results to verify whether or not the measurements are following the pre-established values.

  1. Group related expresses with more emphasis the difference with the proposed approach.

It is a simple integration

  1. The authors claim “in section three, we present the different...”, but the collaborative robots and computer vision systems are all in Section 2.

I didn't understand well, but in section 2 we presente diferent theoretical aspects for inspection and in section 3 the presentation of the developed prototype.

  1. What is the novelty of your paper?

Line 289-292

In the proposed application, the use of collaborative robotic systems allows an excellent interaction between the machines and the personnel who operate with them. In the inspection process, high work rates are not required, which provides a great advantage for collaborative robots over the option of using industrial robots.

  1. The authors should describe the implementation in more detail.

Line 293-297

In this inspection process, the operator can select different parts to be inspected and the use of a rigorous measurement system makes it possible to perform all dimensions in detail at a high inspection speed, guaranteeing high quality and reliability of the inspected part, avoiding errors as well as preventing errors and corrections in the downstream system.

  1. The results of the implementation should be added.

Line 308-312

The system has been in full use in the company for more than 6 months and presents excellent results, in addition to meeting the inspection quality requirements, it is a very easy system to parameterize, which allows the inclusion of new parts for inspection by the operators specialized in this task.

  1. The authors claimed “image processing methods use resources based on depth and surface measurements with high precision. The system fully processes a panel, observing the state of the surface to detect any potential defect in the panels produced to increase the quality of production”, but there are no experimental results to support this conclusion.

This prototype is working in factory in Portugal and they dont give these information the equipment works according to the specifications proposed in the specifications and as the cadence is low the system is robust and functional, in addition it allows the inclusion of new panels in an intuitive way.

Line 306 - 308

The measurements made on the parts to be inspected are validated or not, according to the rejection criteria imposed by the quality system, to guarantee the dimensions requested in the specifications for each object to be inspected.

Reviewer 2 Report

This paper deals properly with an interesting issue. I recommend the publication of the paper after the consideration of the following comments:

1) Since this paper deals with a hot topic, it should include more recent references.

2) The conclusion should highlight the main outcome of the paper and the research limitation.

Author Response

This paper deals properly with an interesting issue. I recommend the publication of the paper after the consideration of the following comments:

  1. Since this paper deals with a hot topic, it should include more recent references.

We made the up-to-date 10 to 16 references

  1. The conclusion should highlight the main outcome of the paper and the research limitation.

In the future, the development of collaborative robotics in the industry will be expressed by the close interaction of man and robot. Under these conditions, the robot's working space intersects with a person's personal space. For an effective interaction between a human operator and a robot, it is necessary to consider production tasks. Industry 4.0 changes the structure of production processes, and man becomes the center of the industrial system. Thanks to the emerging favorable conditions, collaborative robots become significantly brighter, demonstrate the benefits of reliable and secure cooperation, and increase the productivity and efficiency of task execution.

The measurements made on the parts to be inspected are validated or not, according to the rejection criteria imposed by the quality system, to guarantee the dimensions requested in the specifications for each object to be inspected. The system has been in full use in the company for more than 6 months and presents excellent results, in addition to meeting the inspection quality requirements, it is a very easy system to parameterize, which allows the inclusion of new parts for inspection by the operators specialized in this task.

Round 2

Reviewer 1 Report

In the new version of this paper, the authors provide satisfactory replies corresponding  to my questions including revision and the further analysis which highlight the findings and motivations of the manuscript. Thus I think this paper can be accpeted after some improving of the format and readability.